# Incorporating Heterogeneous Features into the Random Subspace Method for Bearing Fault Diagnosis

**DOI:** 10.3390/e25081194

**Published:** 2023-08-11

**Authors:** Yan Chu, Syed Muhammad Ali, Mingfeng Lu, Yanan Zhang

**Affiliations:** 1School of Finance, Shanghai Lixin University of Accounting and Finance, Shanghai 201209, China; 2Department of Engineering Management, National University of Sciences and Technology, Islamabad 44000, Pakistan; syedmali@hfut.edu.cn; 3School of Management, Hefei University of Technology, Hefei 230009, China; lumingfeng@mail.hfut.edu.cn (M.L.); zyn1868@mail.hfut.edu.cn (Y.Z.)

**Keywords:** heterogeneous features, random subspace method, bearing fault diagnosis, deep stack autoencoder, lasso

## Abstract

In bearing fault diagnosis, machine learning methods have been proven effective on the basis of the heterogeneous features extracted from multiple domains, including deep representation features. However, comparatively little research has been performed on fusing these multi-domain heterogeneous features while dealing with the interrelation and redundant problems to precisely discover the bearing faults. Thus, in the current study, a novel diagnostic method, namely the method of incorporating heterogeneous representative features into the random subspace, or IHF-RS, is proposed for accurate bearing fault diagnosis. Primarily, via signal processing methods, statistical features are extracted, and via the deep stack autoencoder (DSAE), deep representation features are acquired. Next, considering the different levels of predictive power of features, a modified lasso method incorporating the random subspace method is introduced to measure the features and produce better base classifiers. Finally, the majority voting strategy is applied to aggregate the outputs of these various base classifiers to enhance the diagnostic performance of the bearing fault. For the proposed method’s validity, two bearing datasets provided by the Case Western Reserve University Bearing Data Center and Paderborn University were utilized for the experiments. The results of the experiment revealed that in bearing fault diagnosis, the proposed method of IHF-RS can be successfully utilized.

## 1. Introduction

Rotating machinery performs an essential function in manufacturing. As a critical element of rotating machines, the bearing often works in harsh environments and can affect the entire machinery’s operation [1,2]. Uncertain faults that occur during bearing performance affect the reliability and safety of the machine, as well as resulting in massive financial losses and fatalities [3]. Therefore, fault diagnosis in rolling bearings has become a significant area of study in modern industries.

In recent times, various fault diagnosis methods centered on machine learning have been proposed for the determination of bearing faults [4]. Feature extraction and fault pattern recognition are two common and fundamental processes for bearing fault diagnosis. During the feature extraction process, features in different domains, like the time domain, frequency domain, and time–frequency domain, have been utilized to enhance the fault diagnosis performance [5,6]. Time domain features can be conveniently extracted by applying statistical calculations, including the mean, variance, standard deviation, etc. [7]. They are suitable for fault diagnosis as well as feature extraction from stationary signals. The time domain features might demonstrate vulnerability to data distinction; in addition, they possess non-linearity, which may cause further difficulties in diagnosis in real applications [8]. Subsequently, frequency domain techniques are taken as alternative choices to describe fault patterns in another respect, as they have a better ability to discover and separate the frequency components. In this class, the most extensively utilized technique is FFT, i.e., fast Fourier transform [9,10]. Thus, in the frequency domain, some features, including the root variance of frequency, the frequency root mean square, and the frequency center, have been extracted by FFT and engaged in bearing fault diagnosis. However, in bearing fault diagnosis using the above methods, the major constraint is their inability to manage non-stationary signals [11]. Moreover, features examining signals in both the time and the frequency domains are known as time–frequency features, and they are viewed as a potent practice for investigating non-stationary signals [8]. Short-time Fourier transform, empirical mode decomposition (EMD), and wavelet packet transform (WPT) are three commonly applied methods for extracting time–frequency domain features that have been used in previous studies [12]. All the features can reflect faults in different aspects and contribute to the final fault diagnosis results. Therefore, appropriate feature extraction approaches and manual feature strategies are required to obtain these statistical features, which require further expertise and domain knowledge. However, through signal processing methods, the extraction of statistical features includes merely superficial information about fault patterns, thus limiting the fault diagnosis performance [13]. To better represent the fault patterns, deeper information about the faults should be considered in the feature extraction process. Deep learning methods can capture more hidden knowledge within hierarchical structures [14,15]. Generally, in bearing fault diagnosis, commonly considered deep learning methods include the convolutional neural network (CNN), long short-term memory network (LSTM), deep belief network (DBN), and stacked auto-encoder (SAE), since deep-learning-based fault diagnosis methods use vibration signals directly as inputs and automatically learn complex diagnostic information from the signals [16,17]. Zhang et al., for instance, proposed a CNN-based network to process two-dimensional image features in an attempt to discover the integral process of the CNN model in feature learning and the classification of fault diagnoses [18]. Further, Qiao et al. developed a dual-input time–frequency model on the basis of a network of LSTM for rolling bearing fault diagnosis, which proved the LSTM method’s effectiveness [19]. Moreover, Shao et al. proposed a unique approach labeled optimization DBN for the bearing diagnosis, whose effectiveness was validated with simulation and experimental signal data [20]. Although these deep learning methods have achieved remarkable diagnosis performance, they still usually require the labeling of information in the learning process since, if the collected labeled data are insufficient, limitations can develop in industrial applications. To address this problem, using an autoencoder (AE) is a better choice since it automatically learns to self-express representations in an unsupervised way. Additionally, by using some stacked AEs, SAEs can extract high-level representational features by setting target values equivalent to the inputs, and, comparably with other networks, they can be conveniently and highly effectively trained. For example, in the SAE network, Liu et al. analyzed the effects of several hidden layers and, in each hidden layer, the neuron number on the model performance [21]. Similarly, Lee et al. mentioned that SAEs can extract highly complex features and, consequently, can be considered more useful for practical applications when using non-linear activation functions [22]. In brief, statistical and deep representation features from diverse perspectives manipulate specific fault information, which also signifies the existence of heterogeneity. However, these heterogeneous features’ complementarity has been rarely explored in bearing fault diagnosis, leaving a large gap regarding the supplementary augmentation of the diagnostic performance. Therefore, combining the novel idea of statistical as well as deep representation features might be a better fusion strategy and a favorable research idea in fault diagnosis. Hence, for conducting a successful bearing fault diagnosis, we adopted such a fusing technique to wholly describe the fault information in this study by combining statistical as well as deep representation features.

During the process of fault pattern recognition, some machine learning methods, such as Artificial Neural Networks (ANNs), Support Vector Machines (SVMs), and Decision Trees (DTs) have been advantageously exploited in the fault diagnosis of bearings [23]. Nonetheless, using a single classification method has some consequences that impact the bearing fault diagnosis performance, like low generalization capability caused by the complicated states of bearing systems [24]. Thus, for dealing with such issues, ensemble learning methods have been utilized, where bearing fault diagnostic decisions are developed from the consensus of several classifiers. Ensemble learning methods can be separated into feature partitioning and instance partitioning methods for the aim of base learner generation. Recently, instance partitioning methods in fault diagnosis, for example, Bagging and Boosting, have been broadly utilized [25,26]. However, the combination of features extracted from different domains will result in a high-dimensional and feature-redundant problem, which may lead Bagging and Boosting methods to perform poorly. Alternatively, fault diagnosis feature partitioning methods, such as random subspace (RS), have proven their superior advantage and capability to cope with the high-dimensional issue [27]. Consequently, on the basis of the above discussion and for the objective of bearing fault diagnosis, the RS method is employed in the present study. Nevertheless, redundant features may be chosen into the same feature subset in RS, leading to the adverse effect on the precision of base learners. Fortunately, one of the sparse methods, the Least Absolute Shrinkage and Selection Operator (lasso) method, can filter out features from high-dimensional feature sets by L1 regularization, improving the prediction performance [28,29]. Benefiting from such excellent performance, this method has been favored in past research. For example, Lateko et al. introduced Lasso into the designed method to achieve effective optimization of learner parameters, and the experimental results confirmed the effectiveness of this method [30]. Duque-Perez et al. improved the traditional Logistic regression classifier with the help of lasso to enhance the model performance of bearing fault diagnosis, and the experimental results confirmed its effectiveness [31]. However, these methods focus more on utilizing lasso to optimize the basic classifier parameters without explicitly incorporating the time domain, frequency domain, and deep representation features related to bearing faults. To overcome these limitations, the RS method and the lasso method are combined in this study to better declare the relationship between multi-domain features and different fault types.

In the current study, a novel random subspace method, i.e., IHF-RS, is proposed by fusing statistical and deep representation features for the precise diagnosis of bearing fault. Firstly, heterogeneous features, including statistical features and deep representation features, are extracted by statistical methods in time domain, frequency domain, and time–frequency domain methods, as well as a deep learning method. Secondly, taking the different predictive power of feature sets into account, a modified lasso is introduced into the RS method for better base classifier construction. Finally, for the purpose of improving the diagnosis accuracy of the bearing fault, a majority voting strategy is employed to aggregate the outputs of various based learners. For verification of the proposed IHF-RS performance, comprehensive experiments are performed on the datasets granted by the bearing data center of Case Western Reserve University (CWRU) and Paderborn University. The experiment results revealed improvements regarding fault diagnosis of bearings via the proposed method, IHF-RS, in comparison with other methods.

The foremost contributions of the current study are summarized as follows:(1)A bearing uncertain breakdown may result in massive financial losses, and an impeccable fault diagnosis is always needed. A framework to enhance bearing fault diagnosis performance is proposed that can fully utilize the heterogeneous features extracted from the bearing vibration data. In this framework, statistical features that are rich in domain knowledge and deep representation features representing high-level non-linear characteristics are incorporated and utilized to further improve the accuracy of bearing fault diagnosis.(2)A novel method for integrating heterogeneous features into a random subspace for conducting a fault diagnosis of bearings is proposed. With such a method, both statistical features and deep representation features are extracted and integrated. Lasso and RS are further combined to handle the problem caused by high-dimensional features. In this way, fault features from different domains can be effectively fused, and the negative impact caused by irrelevant and redundant features can be addressed appropriately.(3)On the CWRU bearing dataset and Paderborn University bearing dataset, empirical studies are performed, and the results attained from the experiments prove that the proposed IHF-RS for bearing fault diagnosis is more effective and viable than other commonly used methods.

The current study is further organized in the following form: In Section 2, the fault diagnosis method, which includes the framework, data acquisition, feature extraction, and model construction, is illustrated. Section 3 explains the experimental design exploiting the CWRU bearing dataset and the Paderborn University bearing dataset. Section 4 depicts the results of the experiments and the discussion. Lastly, a brief study conclusion and future research directions are discussed in Section 5.

## 2. The Proposed Bearing Fault Diagnosis Method

### 2.1. Framework

In modern industries, the bearing is one of the most imperative elements of rotating machinery. To avoid the possible incidence of bearing fault, it is necessary to check the machine bearings’ condition in advance. As shown in Figure 1, the framework in this study has three subsections:(1)Data acquisition. The bearing’s vibration signal data with various faulty forms are acquired.(2)Feature extraction. Using signal processing methods, statistical features in the time, frequency, and time–frequency domains are extracted. Additionally, further significant deep features are extracted via DSAE.(3)Model construction. To weigh different features, modified lasso is introduced, which can help the RS method develop high-quality feature subsets. Then, to train base classifiers, the feature subsets are used. The final fault diagnosis outcomes are achieved by fusing the outputs of each base learner with majority voting.

In practical cases, the process of implementing fault diagnosis by the model is offline; that is, for newly acquired industrial data, the model directly obtains diagnostic results using the trained parameters, which is in real time. Therefore, the proposed method can achieve real-time fault diagnosis.

### 2.2. Data Acquisition

In this paper, we present data on rolling bearings obtained via a specific data acquisition system to build an appropriate bearing fault diagnosis model. In case of localized fault existence, bearing rolling elements run over the fault periodically and generate a series of impulses. The faulty bearing vibration signal has carrier frequencies that are the mechanical structure resonance frequencies besides the fault site. The reciprocal of the period between the impulses is known as modulating frequency, and healthy bearing signals do not have this modulation. The fault in the bearing can be discovered by examining the modulation.

### 2.3. Feature Extraction

The fault existence in machinery parts, such as bearings, can hardly be classified from the raw signal. Based on raw vibration signal data, time domain, frequency domain, time–frequency domain, and deep domain features are extracted in this study.

#### 2.3.1. Time Domain Features

As an initial linear analysis method, time domain analysis can examine the characteristics and structure information of the signals. Time domain features have adequate information regarding a fault, which provides a basic description of the bearing condition. Hence, as presented below in Table 1 and Table 2, in this study, some time domain features are carried out, including square root of amplitude (SRA), root mean square (RMS), shape factor (SF), impulse factor (IF), skewness value (SV), etc. [7,32]. In the table, $x_{i}$ means the *i-th* vibration signal value in the vibration signal sequence, and *N* is the length of the vibration signal sequence.

#### 2.3.2. Frequency Domain Features

Frequency domain features explore lots of valuable information that cannot be identified by time domain features [33]. Previously studies have broadly utilized FFT, i.e., Fast Fourier Transform, to transform the signals in the time domain to the frequency domain. Thus, frequency domain features based on FFT are utilized in this paper. For instance, kurtosis factor of frequency (KFF), skewness factor of frequency (SKFF), kurtosis value of frequency (KVF), skewness value of frequency (SVF), mean of frequency (MEANF), minimum of frequency (MINF), and maximum of frequency (MAXF) are extracted in this paper. The details are shown below in Table 3 and Table 4, wherein $y_{l}$ is the *l-th* vibration signal value in the vibration signal sequence, and *L* shows the length of the vibration signal sequence.

#### 2.3.3. Time–Frequency Domain Features

Signals produced by the momentary vibrations from rolling bearings are always non-stationary due to structural faults. In indicating time-varying signals, the time–frequency approach is a useful method as it is more appropriate for non-stationary signals, as time or frequency domain analysis alone is not enough to provide thorough information on these particular signals [34]. Based on signal decomposition, different kinds of Time–Frequency Analysis (TFA) have previously been employed to examine bearings conditions, such as wavelet package transforms (WPTs), short-time Fourier transform (STFT), and empirical mode decomposition (EMD) [12,35,36,37]. WPT has the potential to find defect-induced transient elements entrenched inside the vibration signal of the bearing, showing its strength in discriminative feature extraction [38,39]. Thus, WPT is adopted to extract time–frequency features in this paper.

Usually, inside several particular frequency bands, fault impulses will be assembled and show fault features along with local energy absorption. So, for feature learning, these informative sub-bands are significant. By calculating the average energy distribution for each sub-band, the fault-relevant frequency band can be chosen. From the original vibration signals, using the WPT $2^{j}$, final leaves of wavelets can be attained with j tree depth. The node energies of $2^{j}$ final wavelet packets are computed and normalized as the time–frequency domain features.

#### 2.3.4. Deep Stack Autoencoder-Based Features

AE method is valuable in locating the raw data representative features, as it can diminish vibration data dimension well and mine the hidden information from high-dimensional features [40]. Normally, an autoencoder is a sort of unsupervised learning and includes three layers. The main process of an AE is illustrated in Figure 2 and Figure 3; the input layer, output layer, and hidden layer are portrayed.

The SAE is generally attained by stacking some AEs [41,42]. In detail, an encoder and a decoder exist in an AE network [43]. The encoder maps the input data into a hidden representation, whereas the decoder reconstructs input data from hidden representations. Specifically, the unlabeled input dataset $X\in {\;R}^{N}$. The encoder layer can squeeze *X* into the representative feature $Y\in R^{M}$(*M < N*); furthermore, the function is exploited as follows:(1)$$Y=f(W^{\left(1\right)}X+b^{\left(1\right)})$$
where *W*^(1)^ and *b*^(1)^ are size M×N weight matrix and bias vector of size *M,* correspondingly. *f*(*x*) is the activation function of AE network. Subsequently, representative feature Y is rebuilt into the vector $\hat{X}$ by the decoder layer in this manner:(2)$$\hat{X}=f(W^{\left(2\right)}Y+b^{\left(2\right)})$$
where *W*^(2)^ along with *b*^(2)^ are prescribed similarly as *W*^(1)^ and *b*^(1)^, correspondingly. The key objective of AE network training is to obtain $\theta =\{W^{\left(1\right)},W^{\left(2\right)},b^{\left(1\right)},b^{\left(2\right)}\}$ by decreasing reconstruction error involving X and $\hat{X}$. From the stacked encoding layers, deep representation features can be acquired once all layers have been trained.

### 2.4. Model Construction

One of the significant phases in bearing diagnosis for fault identification is that high-dimensional features extracted from raw signals in multi-domains often include unnecessary interconnected components. The base classifiers’ performance is negatively influenced by this, which may cause an inappropriate diagnostic output of the ensemble methods in the future. The feature subsets are randomly selected in the RS method instead of utilizing the complete feature set to assure the base classifiers’ differences [44]. Although there will be more chances of selecting relevant and redundant features in case the feature sampling process is fully random, this will result in generating deprived base classifiers, which will further result in a deprived ensemble. So, in the feature subset construction phase, it is essential to regard the significance of each feature to make sure that, with greater probability, significant features can be chosen compared to redundant features. To choose high-quality feature subsets for training each base classifier, a semi-random subspace technique is presented in this paper. Hence, this will enhance the accuracy of base classifiers while conserving their diversity at the same time. For attaining this objective, lasso entailing a contraction estimation method is exploited in this study to make enhancements to the RS method. Specifically, the lasso method is employed to adaptively select reliable features for different feature subspaces. In this way, the process of feature selection is embedded in the optimization process, which means that for the proposed method, the selection of reliable features is adaptively selected based on the optimization objective. Tibshirani suggested the lasso method, which assists in the enhancement of classification performance by obtaining a sparse way out of high-dimensional feature space and concurrently avoiding overfitting [28]. The class label and features are taken as dependent and independent variables, and afterward, every single variable coefficient is calculated using linear regression. With penalty parameter adjustments, several coefficients can shrivel to zero, and more correlated features for the class label can be recognized and chosen. Unlike the conventional feature selection method, which assesses every single feature in isolation, in the model selection process, the lasso estimation shrinks the overall universal feature space [29].

In Figure 4, the process of the proposed method is presented, based on three major steps. The initial step is about feature subspace generation. The next step is base classifier training for the ensemble. The final step is the combination of different base learner outputs.

Initially, in the first step, a predefined number of feature subsets are formed after splitting the original feature set by regulating significant parameters, i.e., penalty parameter λ and subspace rate r of lasso estimation. Where penalty parameter λ influences the feature set shrinkage, r decides each partitioned feature subset ratio to the universal feature set. Using lasso estimation, the weight of each feature is decided. The theory is further explained in this way. For obtaining the model, the addition of the squared residuals should be reduced by lasso, utilizing summation of the regression coefficients to constrain the absolute value to be smaller than a constant.

In a given set of data $D={\left\{(x_{1},y_{1}),\ldots ,(x_{i},y_{i}),\ldots ,(x_{n},y_{n})\right\}}^{T}$, where vector space pattern is stated as $x_{i}\in R^{m}$, $x_{i}=\{x_{1,i},x_{2,i},\ldots ,x_{c,i},\ldots x_{m,i}\}$ means features, m is feature number, and the label is represented by $y_{i}$, whereas the number of instances is denoted by n. Since in the regression setup, the observations are independent, or the labels are independently and provisionally given c the feature $x_{c,i}$. Furthermore, $x_{c,i}$ can be standardized as $\frac{1}{n}\sum_{i=1}^{n}x_{c,i}=0,\frac{1}{n}\sum_{i=1}^{n}x_{c,i}^{2}=1$. Accordingly, the lasso estimate is defined by the following:(3)$$\arg\underset{\gamma }{\min}\{\sum_{i=1}^{n}(y_{i}-\sum_{c=1}^{m}{\left(\gamma_{c}x_{c,i}\right)}^{2}\}+\lambda \sum_{c=1}^{m}\left|\gamma_{c}\right|$$

When the λ value is suitably large, so highly correlated variables can be discovered and retained, it will root the shrinkage of the solution to 0, with several coefficients probably equivalent absolutely to 0. Here, $\gamma_{c}$ defines the regression coefficient of the feature $x_{c}$, and λ defines penalty parameter, which administers the shrinkage degree. Global feature set T series will be generated once λ value is comparatively smaller, and its limitation will be eradicated. Subsequently, the determination of feature weight can be illustrated as follows: Initially, degree of correlation between $x_{c}$ feature and class label $y_{i}$ is produced by employing lasso estimation. Afterward, a set of significant scores can be given for each feature, which can be represented as $\left\{\gamma_{1},\gamma_{2},\ldots \gamma_{c},\ldots \gamma_{m}\right\}$. Moreover, the weight of the feature is then decided by the following:(4)$$W_{c}=\frac{\left|\gamma_{c}\right|}{\sum_{c=1}^{{}^{m}}\left|\gamma_{c}\right|}$$

After the feature weight determination, from the original set of data D, S sub-datasets are randomly generated. For the feature $x_{c}$, once the feature weight $w_{c}$ is obtained, it can be randomly extracted with its feature weight and the subspace rate r. The weights of all features can be represented as ${w=\{w}_{1}{,w}_{2},\cdots ,w_{c},\cdots ,w_{m}\}$. Let us assume number of subspaces to be s, and semi-random feature subspace can be depicted like $L_{sub}^{j}=\{(x_{1}^{j},y_{1}),\ldots ,(x_{i}^{j},y_{i}),\ldots ,(x_{n}^{j},y_{n})\},1\leq j\leq s$. A set regarding feature subspaces $\left\{L_{sub}^{1},L_{sub}^{2},\ldots ,L_{sub}^{j},\ldots ,L_{sub}^{s}\right\}$ can be attained by repetitively extracting the features for every subspace. The key significant features that have greater probability can be extracted from the feature subset. Additionally, by randomly selecting the features, assortment of the base learners is raised. Thus, classification accuracy can be notably enhanced. Thereafter, in the next step, based on the sub-datasets construction, selected base learners are trained. In this study, SVM is chosen as a base learner since it has been verified as the best classifier in bearing fault diagnosis [25,45,46]. Moreover, in complex classification models, SVM works better and has the quality of handling non-linear data. Normally, SVM minimizes generalization error by minimizing structural risk. In a high-dimensional feature space, non-linear input vectors in SVM with a kernel function are mapped.

Given training set instances $D={\left\{(x_{1},y_{1}),\ldots ,(x_{i},y_{i}),\ldots ,(x_{n},y_{n})\right\}}^{T}$ and $x_{i}=\{x_{1,i},x_{2,i},\ldots ,x_{c,i},\ldots ,x_{m,i}\}$, here, feature dimension size is denoted by m, and i∈{1,2,⋯,n} indicates vector space pattern. For minimizing the probable rate of misclassification, SVM strives to seek a hyperplane linear classifier f(x), characterized as $f(x)=\mathrm{sgn}(w^{T}x+b)$. In SVM, looking for the most favorable classifier f(x) is similar to finding solution for a convex quadratic optimization problem:(5)$$\begin{array}{l} \underset{w,b}{\max}\frac{1}{2}{\left\|w\right\|}^{2}+C\sum_{i=1}^{n}\xi_{i}\\ subject\;to\;y_{i}(\langle w,x_{i}\rangle +b)\geq 1-\xi_{i}(\xi_{i}\geq 0,i=1,\ldots ,n) \end{array}$$
where *C* indicates regularization parameter. On the training set D, it is utilized to stabilize classifier’s complications and classification accuracy. Above quadratic problem is usually answered via its twofold conception. With a non-linear kernel function shift of the engaged vector inner-product, linear SVM can be transformed into further alterable non-linear SVM. Ordinary kernel functions are comprised of polynomial, linear, sigmoid, and radial bases.

Further, in the third step, the objective is to cumulate each base learner classification result to reduce the classification errors. Many researchers utilized the majority voting rule, which is considered a useful aggregation method. Based on its advantages, in this study, we also applied it to cumulate the base learner results.

Given a base learner set $\left\{C_{i}(x),1\leq i\leq S\right\}$, the majority voting rule is expressed as follows:(6)$$C^{*}(x)=\mathrm{sgn}\{\sum_{i}C_{i}(x)-\frac{S-1}{2}\}$$

The pseudo-code of IHF-RS algorithm is presented below (Algorithm 1):

**Algorithm 1.** Pseudo-code of IHF-RS algorithm.

$Semi-RS(D,\lambda ,r,S,L)\;Input:Dataset\;D=\left\{\left(x_{1},y_{1}\right),\left(x_{2},y_{2}\right),\ldots ,\left(x_{n},y_{n}\right)\right\}\;\begin{array}{l} \mathrm{Lasso penalty parameter \lambda ;}\\ \;\;\;\;\mathrm{Random subspace rate}\;r;\\ \;\;\;\;\mathrm{Baseleaner number}\;S;\\ \;\;\;\;\mathrm{Baseleaner}\;L.\\ \mathrm{Output}:H(x) \end{array}\;\begin{array}{l} Processing:\\ for\;c\in \{1,2,\ldots ,m\}\;do\\ \;\;\;\;\gamma_{c}=grouplasso(D,\lambda )\;\\ \;\;\;{\;w}_{c}=\frac{\left|\gamma_{c}\right|}{\Sigma_{c=1}^{m}\left|\gamma_{c}\right|}\;\\ end\;for\\ for\;s\in \{1,2,\ldots ,S\}\;do\\ \;\;\;{\;D}_{s}=\mathrm{RS}(D,r,w)\;\\ \;\;\;{\;h}_{s}=L\left(D_{s}\right)\\ end\;for\\ H(x)={\mathrm{argmax}}_{y\in Y}\Sigma_{s=1}^{S}1\left(y=h_{s}(x)\right) \end{array}$



## 3. Experimental Design

### 3.1. Experimental Dataset

For the validation of the proposed method in the current paper, two signal datasets regarding bearing vibration given by the CWRU Bearing Data Center and Paderborn University were utilized. For the CWRU bearing dataset, the dataset was obtained with bearing accelerometer sensors during multiple bearing conditions and functional loads. The test rig apparatus is given in Figure 5, which was utilized to obtain the vibration data with the help of an electric motor, a torque transducer/encoder, and a dynamometer. For testing purposes, three sorts of bearing faults, i.e., outer race fault, ball fault, and inner race fault, from a diameter of 0.007 to 0.028 inches, were introduced by an electro-discharge apparatus. From healthy and faulty bearings, the vibration signals were obtained on the test rig at 12 kHz and 48 kHz sampling frequencies around 10 s. The test rig functioned with four distinct loads of 0, 1, 2, and 3 hp at a speed of 1797–1720 Rpm. More comprehensive details concerning the test set can be found in [47]. For the Paderborn bearing dataset, the dataset was given by Christian Lessmeier from Paderborn University. The test rig consists of five key components, such as the electric motor, flywheel, testing module, measurement shaft, and load motor, which are shown in Figure 6. In this dataset, 6 normal bearing sets as well as 26 damaged bearing sets are collected, plus both the vibration signal and the current signal were collected for 4 s at 64 kHz. The details of the dataset can be found in [48]. In this study, only six sets of them, including the inner fault bearing set, the outer fault bearing set, and the normal sample set were selected.

For evaluating the performance of the proposed methods in this paper, the CWRU bearing vibration dataset was divided into four subsets that are symbolized with VD_0, VD_1, VD_2, and VD_3. Using drive-end bearings, these specific subsets are obtained at a sampling frequency of 48 kHz during four dissimilar motor loads of 0 hp, 1 hp, 2 hp, and 3 hp, respectively. Ten dissimilar bearing states are simulated in the present study, comprising a regular condition, a ball fault (BF), an outer fault (OF), and an inner fault (IF). The complete signals are split into uninterrupted intervals every 1024 points without any overlapping. The purpose was to feed classifiers by generating more instances. In Table 5, more details about the experimental datasets are mentioned, in which “007”, “014”, and “021” show that the diameters of the faults are 0.007, 0.014, and 0.021 inches. In the Paderborn bearing dataset (Pdata), three inner fault types of samples, two outer fault types of samples, and the normal samples with 64 kHz resolution are selected from the dataset every 1024 points without overlap. To verify the proposed scheme, the dataset is divided into training datasets (90%) and test datasets (10%).

From instances of 10 dissimilar fault types in VD_3 of the CWRU bearing dataset, Figure 7 depicts waveforms related to the time domain along with their subsequent frequency spectra. Even so, because of the original vibration signals’ incredibly high dimensionality, it is further required to process and calculate the signal features. Based on the proposed method, from the time domain (F1), 16 features are extracted, whereas from the frequency domain (F2), 12 features are extracted. While 2^5^ features are attained by utilizing WPT with the mother wavelet for time–frequency domain features (F3) for breaking the original signals down at the fifth level. Thus, from the time, frequency, and time–frequency domains, the total extracted features by the signal processing methods are 60. The deep representation features extracted through DSAE (F4) are 64. Specifically, the layer number of the SDAE is 7, and the detailed network parameter settings are 1024, 700, 300, 64, 300, 700, and 1024. Adam is selected as the optimizer with a learning rate of 0.001, and the batch size is set to 256, training for 200 batches. Meanwhile, Dropout and Batch Normalization are adopted to defeat the overfitting problem during deep representation feature extraction.

### 3.2. Performance Evaluation Criteria

In this paper, for evaluating the performance of the proposed method, a commonly used metric, i.e., average accuracy (ACC), is utilized. For a sample to be classified with a given classifier, four different types of conceivable results exist, which are True Positive (*TP*), False Positive (*FP*), True Negative (*TN*), and False Negative (*FN*). In these results, bearing faulty instances can be treated as a positive class and the others as a negative class. The accuracy of commonly used indicators is defined as follows:(7)$$AverageAccuracy=\frac{TP+TN}{TP+FP+FN+TN}$$

### 3.3. Compared Methods

In our experiments, the given proposed method IHF-RS is compared to the SVM, MLP, and four other popular ensemble methods. These methods are Bagging, Adaboost, and the standard RS method. For a valid comparison, we set the base learners of Bagging, Adaboost, and the RS method as the SVM. The rate of the RS method and the penalty parameter of *l*_2,1_ norm regularization are significant parameters of this method. The details about the parameters used in the experiments are listed in Table 6. It must be noted that the features that can achieve optimal accuracy are considered reliable features of the compared models and are adopted.

### 3.4. Experimental Procedure

For verification of the proposed method, IHF-RS, all the comparative experiments are conducted ten times with 10-fold cross-validation, for a total of one hundred experiments. In a cross-validation, nine folds are taken for training, whereas the remaining fold is left for testing. For classifying the testing set, the highest average accuracy parameter settings selected from training were selected. During this process, we ensured that the distribution of the training data was the same as that of the test data, which means that the training data covered all possible types of faults. The number of selected features was fixed throughout the entire process. By calculating the classification accuracy and mean of these 100 experiments, the ultimate results were obtained, which makes the results statistically sound. In RS, the regularization coefficient and learning rate are imperative parameters. The experimental flow is illustrated in Figure 8. The proposed method is fully capable of generalizing it to solve multiple fault diagnosis tasks. Specifically, by combining the lasso method and the RS method, we can adaptively select reliable fault features from multi-domains, and train multiple basic classifiers with multiple fault data to obtain their respective multi-classification results. Finally, we can use the major voting mechanism to achieve integration, thereby obtaining accurate multiple fault recognition results.

### 3.5. Experimental Results

The mean accuracy calculated by using the result of tenfold cross-validation with 10 times the methods is chosen as the evaluation criteria. The mean accuracy of the proposed and comparison methods is presented in Table 7, and the best results are highlighted. From Table 5, it is depicted that practically every single mean accuracy optimum result is performed by the proposed method, i.e., 98.37% (VD_0), 96.30% (VD_1), 95.95% (VD_2), 95.83% (VD_3), and 98.51% (Pdata), respectively. Meanwhile, it is clearly visible that these results are better than the other methods compared. Bearing fault diagnosis method performance degradation occurred due to the increased load of the bearing system, and the possible reason for this is that an increasing load makes the test rig highly complex [49]. Consequently, under such operating situations, the datasets may comprise some noisy data. Accordingly, the proposed method demonstrates better steadiness compared to the other methods. Moreover, the Adaboost method exhibits poorer performance than other ensemble methods. Thus, it is clearly sensible to assert that there exists an overfitting issue in the Adaboost training process that is caused by the noise instances. Hence, incorporating heterogeneous features improves the prediction accuracy, and the proposed method is appropriate for implementation. In brief, the results based on the experiment revealed that our proposed method can be lucratively utilized for fault diagnosis of bearing.

## 4. Model Analysis

### 4.1. Evaluation of the Incorporated Features

For assessing the significance and usefulness of diversified feature subsets, a correlation analysis was carried out using features from the time domain, frequency domain, time–frequency domain, and DSAE. For verifying the effectiveness of different feature subsets, Figure 9 illustrates the classification accuracies of different domain features. These include time domain features (F1), frequency domain features (F2), time–frequency domain features extracted by WPT (F3), deep representative features extracted by DSAE (F4), and their combinations [50]. Furthermore, it is visible from Figure 10, that in statistical features, a resilient internal correlation exists, while in deep representation features extracted by DSAE, it is low. This reveals the effectiveness of the DSAE approach in coping with redundant and interrelated features. Regarding distinct features, it is depicted in Figure 10, that from time domain and frequency domain features, the prediction results of time–frequency domain features are almost better. Besides, compared with the statistical features, the prediction results produced by deep representation features are preeminent. From the accuracies of different datasets, it can be comprehended further that the increasing bearing system load results in bearing fault diagnosis methods performance degradation. Such as the time domain feature rates of 5.90%, 17.52%, and 22.52%, and the frequency domain feature prediction accuracy in inconsistent loads of 16.43%, 18.32%, and 22.11%, comparatively lower than VD_0. During these operating circumstances, some noisy data may prevail in the dataset. However, deep learning features reduced prediction accuracy by 6.27%, 8.63%, and 2.56% in comparison with VD_0. Comparatively, with the statistical features, the information expression enhancement degree is considerably impacted by the noise. Therefore, deep representation feature extraction of DSAE features is steadier, and the information description is extra thorough and inclusive.

Additionally, according to Figure 10, most of the methods have attained optimum accuracies under combined features, which shows that they can further improve each other’s performance and complement each other. In VD_0, the average performance of combined features is improved by 8.34% compared with F1, 8.81% compared with F2, 2.79% compared with F3, and 2.35% compared with F4. In bearing fault diagnosis, such significant enhancements in accuracies confirm and verify the usefulness of the fusing features. Moreover, compared with the usual random subspace, each dataset’s performance is improved with the method by 0.85% (VD_0), 0.53% (VD_1), 0.71% (VD_2), 0.73% (VD_3), and 0.48% (Pdata), respectively. However, not all methods are appropriate for combined feature prediction. The prediction results of Adaboost on combined features declined as compared to the deep representative features, either because of the noise or the overly large feature dimensions. Overall, the combination of features has a positive effect on bearing fault diagnosis, and the proposed method can rationally solve the correlation and redundancy issues among features.

### 4.2. Evaluation of the Parameter

The proposed method has a superlative diagnostic effect prediction for combined features, but its performance fluctuates in various parameters. In the current study, the learning rate parameter is selected, whose influence on the accuracy is shown in Figure 11. From the view of the following datasets, the proposed method attained preeminent accuracy with *ratio =* 0.5 on VD_0, *ratio =* 0.7 on VD_1, *ratio =* 0.7 on VD_2, *ratio =* 0.7 on VD_3, and *ratio =* 0.7 on Pdata. It can be observed that the performance of the proposed method gradually rises and then falls with different *ratio* values ranging from 0.1 to 0.9. This is a sign that indicates that redundant features may be present in the original feature space [51]. Likewise, the highest mean accuracy was achieved with *ratio* values equal to 0.5 and 0.7. The reason is that the import features are selected first by the structured sparsity learning model in this method. It is not easy to identify the exact optimum values of this parameter, as different optimum values are obtained on different datasets. Yet, it is clear that the performance of the proposed method is probably affected by the ratio. Generally, it can be summarized that the proposed method’s effectiveness for bearing fault diagnosis can be significantly enhanced if the engaged parameters are tuned properly.

### 4.3. Confusion Matrix

To further validate the effective performance of the proposed method in bearing fault diagnosis, we visualized the diagnostic results of the proposed method and the comparison methods on the Pdata dataset. The details of the confusion matrix are shown in Figure 12. It can be observed that the proposed method has superior performance in fault recognition accuracy for various categories compared to other comparative methods. In addition, in terms of identifying two types of outer faults, the proposed method is significantly superior to other methods. The reason should be that the two types of fault patterns are relatively similar, and it is necessary to fully integrate multi-domain features to achieve accurate differentiation.

## 5. Conclusions and Future Research Directions

For enhancing the mechanical system’s performance and reliability in rotary machinery, the diagnosis of faults in the rolling component bearing is very essential, since the failure of bearings is one of the most recurring reasons for breakdowns in rotary machinery. Thus, a novel approach that incorporates heterogeneous features into the random subspace method is suggested for bearing fault diagnosis in the present study. In this suggested method, both statistical features and DSAE-based deep representative features are extracted. Then, a modified lasso that can guide the feature fusion is introduced in the RS method to handle the issue of high dimensionality and further enhance the performance of the fault diagnosis. For substantiating the method’s efficacy assimilated with existing methods, experiments are conducted on the CWRU bearing dataset and the Paderborn University bearing dataset. It is also further revealed that the proposed method adeptly attains finer accuracies, illustrating the superiority of the proposed method in bearing fault diagnosis.

It’s vital to state that the proposed method has resulted in positive results with improved accuracy, even though some further directions for future research exist. Firstly, in this paper, the proposed framework requires verification on vast and assorted bearing datasets to validate the generalization performance further. Secondly, although the lasso is introduced in the semi-random subspace method in this paper, other suitable methods can also be used for effective feature subspace construction. Thirdly, to cope with the high-dimensional problem, as the proposed method is intensive computationally, parallel computing methods need to be further discovered to solve such difficulties in future studies. Fourth, we will further explore the situation where training data only contains partial fault-type data in future research work and combine more advanced technology to solve this problem. More advanced technologies will be considered to solve this problem, such as transfer learning.

## Figures and Tables

**Figure 1 entropy-25-01194-f001:**
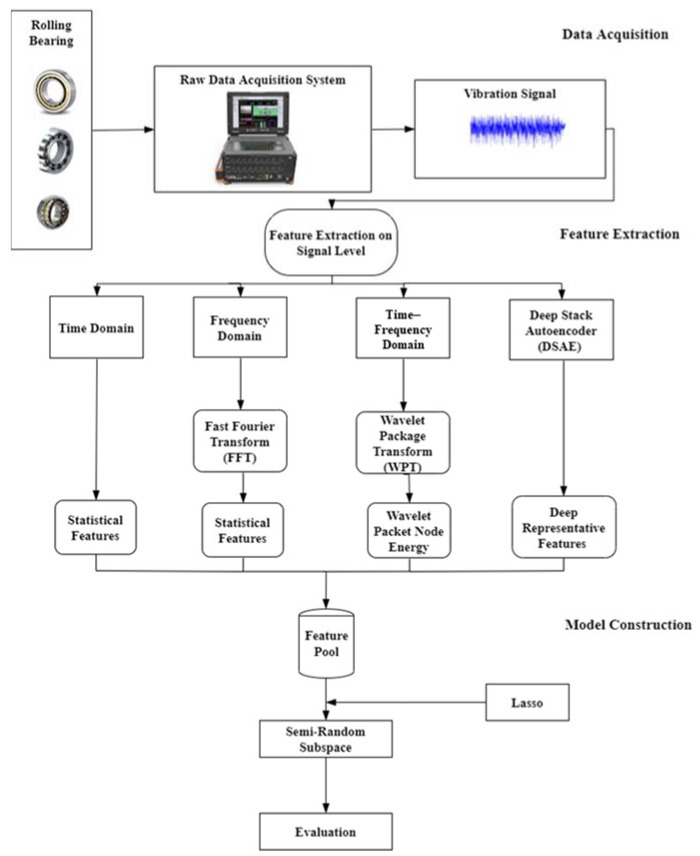
Proposed method framework.

**Figure 2 entropy-25-01194-f002:**
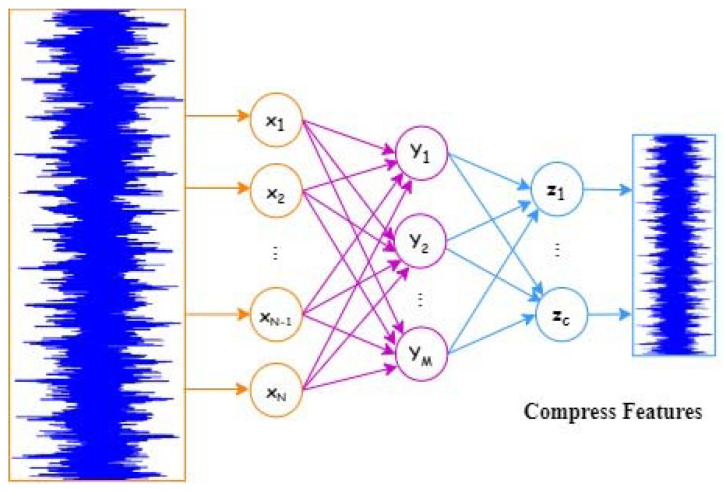
SAE main process.

**Figure 3 entropy-25-01194-f003:**
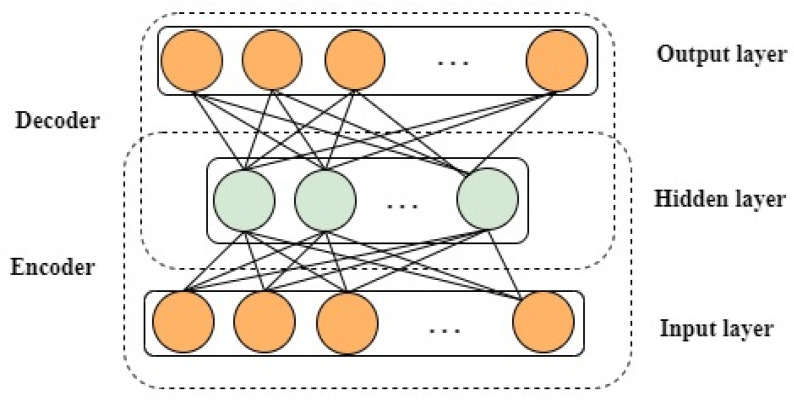
Structure of autoencoder.

**Figure 4 entropy-25-01194-f004:**
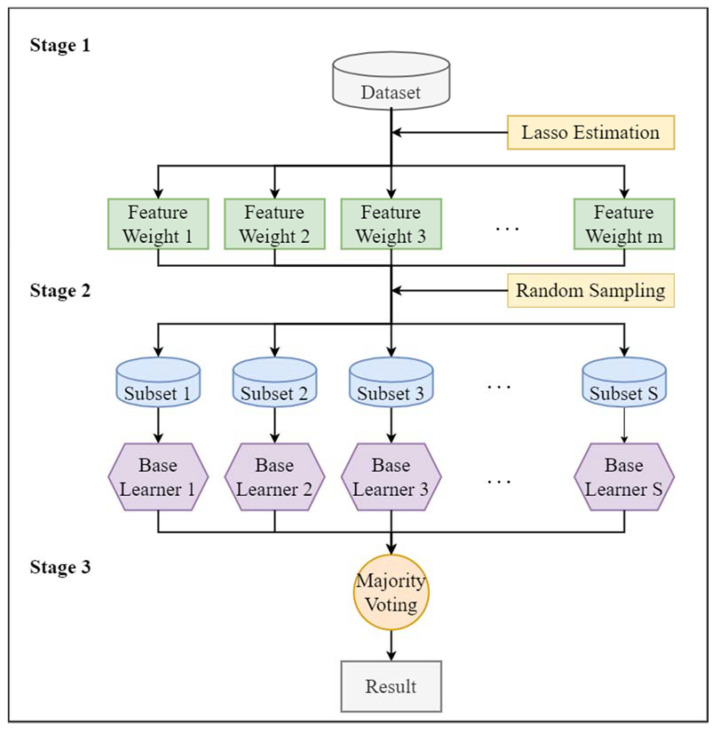
Process of the proposed method.

**Figure 5 entropy-25-01194-f005:**
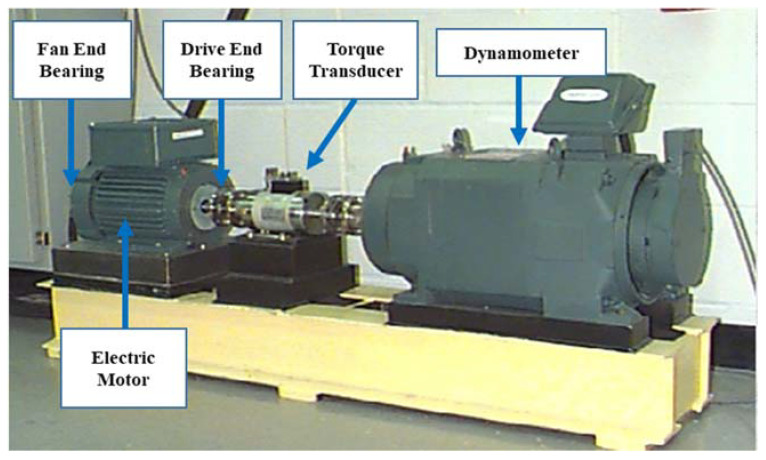
Test rig of the CWRU.

**Figure 6 entropy-25-01194-f006:**
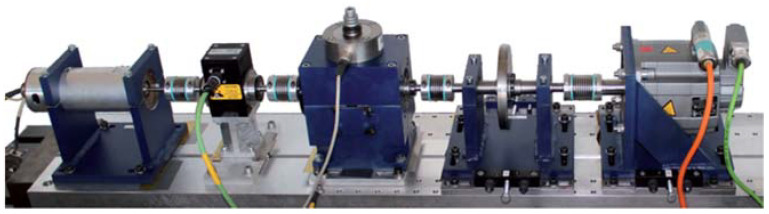
Paderborn dataset test rig.

**Figure 7 entropy-25-01194-f007:**
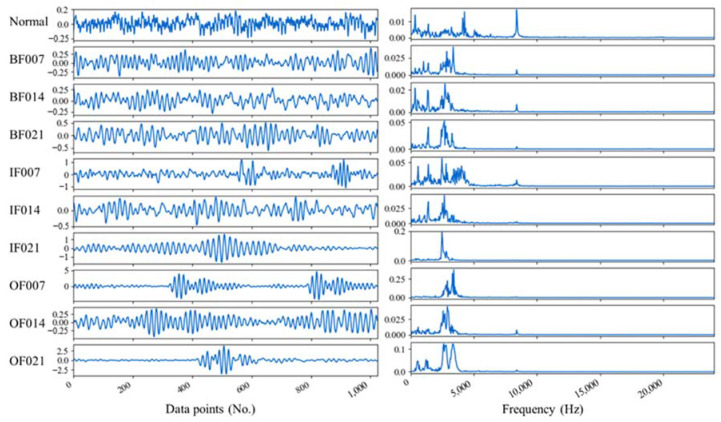
(**Left**) Time domain waveforms in different conditions and (**right**) their corresponding frequency spectra.

**Figure 8 entropy-25-01194-f008:**
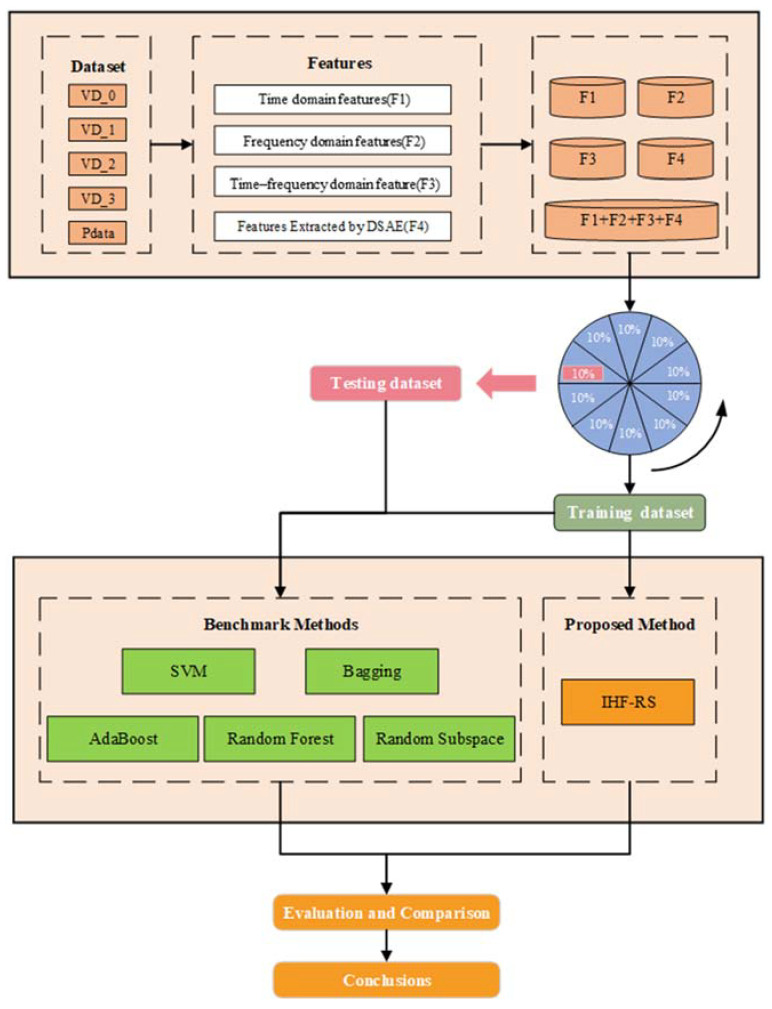
Procedure of the Experiment.

**Figure 9 entropy-25-01194-f009:**
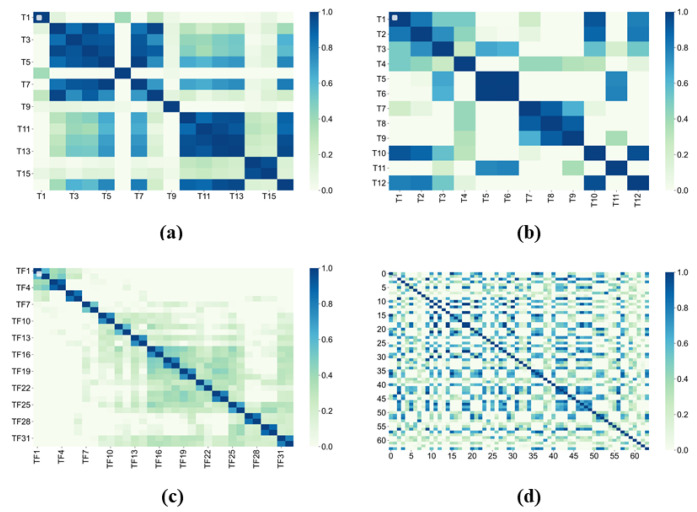
Feature correlation analysis on four feature subsets in the VD_0 dataset: (**a**) time domain features; (**b**) frequency domain features; (**c**) time–frequency domain features; (**d**) DSAE-based deep representative features.

**Figure 10 entropy-25-01194-f010:**
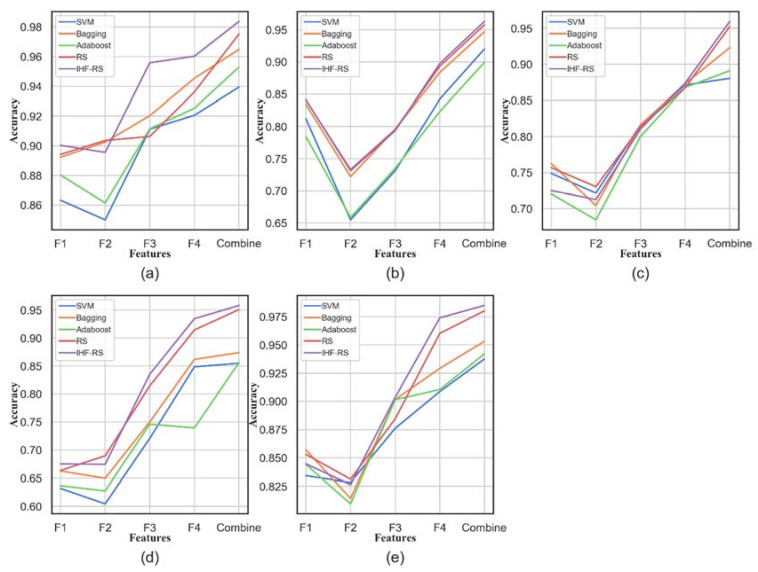
Different features’ accuracy comparisons and their fusion on four datasets: (**a**) VD_0; (**b**) VD_1; (**c**) VD_2; (**d**) VD_3; (**e**) Pdata.

**Figure 11 entropy-25-01194-f011:**
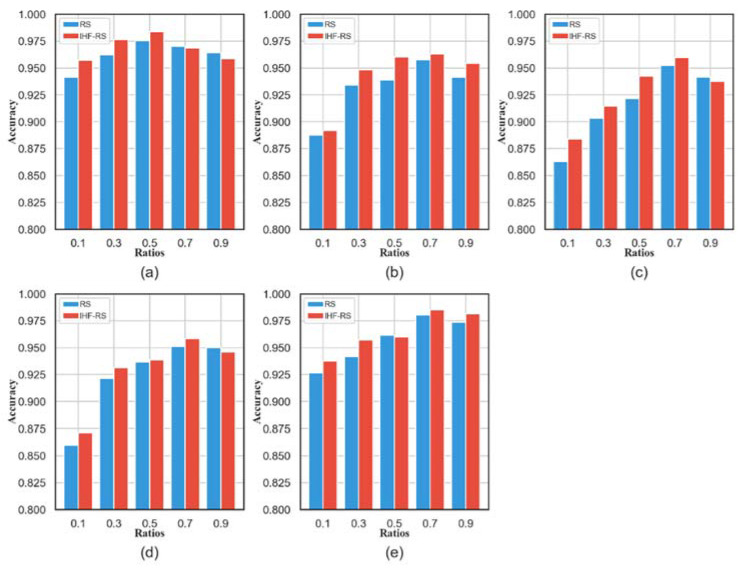
Sensitive analysis of accuracy for the proposed method and RS on four datasets: (**a**) VD_0; (**b**) VD_1; (**c**) VD_2; (**d**) VD_3; (**e**) Pdata.

**Figure 12 entropy-25-01194-f012:**
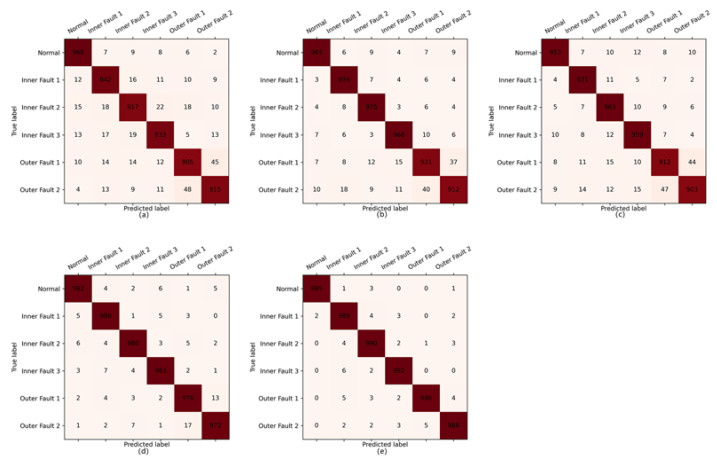
Confusion matrix of different methods on Pdata datasets: (**a**) SVM; (**b**) Bagging; (**c**) Adaboost; (**d**) random subspace; (**e**) IHF-RS.

**Table 1 entropy-25-01194-t001:** Time domain feature definitions.

Formula	Formula
$X_{mean}=\frac{1}{N}\sum_{i=1}^{N}x_{i}^{}$	$X_{mav}=\frac{1}{N}\sum_{i=1}^{N}\left|x_{i}^{}\right|$
$X_{rv}={\left(\frac{1}{N}{\sum_{i=1}^{N}\left(x_{i}-X_{mean}\right)}^{2}\right)}^{1/2}$	$X_{max}=max(x_{i})$
$X_{min}=min(x_{i})$	$X_{rms}={\left(\frac{1}{N}\sum_{i=1}^{N}x_{i}^{2}\right)}^{1/2}$
$X_{sra}={\left(\frac{1}{N}\sum_{i=1}^{N}\sqrt{\left|x_{i}\right|}\right)}^{2}$	$X_{kv}={\left(\frac{1}{N}\sum_{i=1}^{N}\left(\frac{x_{i}-X_{mean}}{X_{rv}}\right)\right)}^{4}$
$X_{sv}={\left(\frac{1}{N}\sum_{i=1}^{N}\left(\frac{x_{i}-X_{mean}}{X_{rv}}\right)\right)}^{3}$	$X_{ppv}=X_{max}-X_{min}$
$X_{cf}=max\left(\left|x_{i}\right|\right)/X_{rms}$	$X_{if}=max\left(\left|x_{i}\right|\right)/X_{abs}$
$X_{mf}=max\left(\left|x_{i}\right|\right)/X_{sra}$	$X_{kf}=X_{kv}/X_{rms}^{4}$
$X_{shf}=X_{rv}/X_{abs}$	$X_{skf}=X_{kv}/X_{rv}^{3}$

**Table 2 entropy-25-01194-t002:** Description of time domain features.

Features	Description	Features	Description
$X_{mean}$	Mean of time domain signals	$X_{mav}$	Mean of absolute values of time domain signals
$X_{rv}$	Standard deviation of time domain signals	$X_{max}$	Maximum value of time domain signal
$X_{min}$	Minimum value of time domain signal	$X_{rms}$	Root mean square of time domain signal
$X_{sra}$	Square root of amplitude of time domain signal	$X_{kv}$	Kurtosis of time domain signal
$X_{sv}$	Skewness value of time domain signal	$X_{ppv}$	Peak-to-peak value of time domain signal
$X_{cf}$	Ratio of maximum absolute value to Mean squared error	$X_{if}$	Ratio of maximum absolute value to absolute value
$X_{mf}$	Ratio of maximum absolute value to square root of amplitude	$X_{kf}$	Ratio of kurtosis to the 4th power of the root mean square
$X_{shf}$	Ratio of standard deviation to absolute value	$X_{skf}$	Ratio of kurtosis to the 3rd power of the standard deviation

**Table 3 entropy-25-01194-t003:** Frequency domain features definitions.

Formula	Formula
$X_{meanf}=\frac{1}{L}\sum_{l=1}^{L}y_{l}$	$X_{rvf}={\left(\frac{1}{L}{\sum_{l=1}^{L}\left(y_{l}-X_{meanf}\right)}^{2}\right)}^{1/2}$
$X_{maxf}=max(y_{l})$	$X_{minf}=min(y_{l})$
$X_{rms}={\left(\frac{1}{L}\sum_{l=1}^{L}y_{l}^{2}\right)}^{1/2}$	$X_{svf}={\left(\frac{1}{L}\sum_{l=1}^{L}\left(\frac{y_{l}-X_{meanf}}{X_{rvf}}\right)\right)}^{3}$
$X_{kvf}={\left(\frac{1}{L}\sum_{l=1}^{L}\left(\frac{y_{l}-X_{meanf}}{X_{rvf}}\right)\right)}^{4}$	$X_{skff}=X_{kvf}/X_{rvf}^{3}$
$X_{kff}=X_{kvf}/X_{rmsf}^{4}$	$X_{fc}=\sum_{l=1}^{L}\left(f_{l}\cdot y_{l}\right)/X_{meanf}$
$X_{rmswf}={\left(\frac{1}{L}\sum_{l=1}^{L}\left(f_{l}^{2}\cdot y_{l}\right)/X_{meanf}\right)}^{1/2}$	$X_{rvwf}={\left(\frac{1}{L}\sum_{l=1}^{L}\left(f_{l}-X_{fc}{)}^{2}\cdot y_{l}\right)/X_{meanf}\right)}^{1/2}$

**Table 4 entropy-25-01194-t004:** Description of frequency domain features.

Features	Description	Features	Description
$X_{meanf}$	Mean of frequency	$X_{rvf}$	Standard deviation of frequency
$X_{maxf}$	Maximum of frequency	$X_{minf}$	Minimum of frequency
$X_{rms}$	Root mean square of frequency	$X_{svf}$	Skewness value of frequency
$X_{kvf}$	Kurtosis value of frequency	$X_{skff}$	Skewness factor of frequency
$X_{kff}$	Kurtosis factor of frequency	$X_{fc}$	Gravity frequency
$X_{rmswf}$	Mean square deviation waveform factor	$X_{rvwf}$	Standard deviation waveform factor

**Table 5 entropy-25-01194-t005:** The Experimental Datasets.

Datasets	Description	Number of Classes	Number of Instances
VD_0	Normal, BF007, BF014, BF021, IF014, IF021, OF007, OF014, OF021	9	9 × 200
VD_1	Normal, BF007, BF014, BF021, IF007, IF014, IF021, OF007, OF014, OF021	10	10 × 200
VD_2	Normal, BF007, BF014, BF021, IF007, IF014, IF021, OF007, OF014, OF021	10	10 × 400
VD_3	Normal, BF007, BF014, BF021, IF007, IF014, IF021, OF007, OF014, OF021	10	10 × 400
Pdata	Normal, Inner Fault 1, Inner Fault 2, Inner Fault 3, Outer Fault 1, Outer Fault 2	6	6 × 1000

**Table 6 entropy-25-01194-t006:** Details of the parameters used in the experiments.

Methods	Parameters
SVM	Kernel: ‘rbf’. Gamma: 1/number of features. Penalty: 1.0.
Bagging	Number of base classifiers: 10. Base classifier: SVM.
Adaboost	Number of base classifiers: 10. Base classifier: SVM.
Random Subspace	Subspace ratio: (0.1, 0.3, 0.5, 0.7, 0.9). Number of base classifiers: 10. Base classifier: SVM.
IHF-RS	Penalty: (0.0001, 0.001, 0.01, 0.1, 1). Subspace ratio: (0.1, 0.3, 0.5, 0.7, 0.9). Number of base classifiers: 10. Base classifier: SVM.

**Table 7 entropy-25-01194-t007:** Comparison methods’ accuracy (mean).

Methods	VD_0	VD_1	VD_2	VD_3	Pdata
SVM	0.9374	0.9200	0.8804	0.8550	0.9376
Bagging	0.9648	0.9469	0.9230	0.8740	0.9533
Adaboost	0.9529	0.8993	0.8909	0.8563	0.9421
Random subspace	0.9752	0.9576	0.9524	0.9510	0.9803
IHF-RS	**0.9837**	**0.9630**	**0.9595**	**0.9583**	**0.9851**

## Data Availability

The datasets used or analyzed during the current study are available from the corresponding author on reasonable request.

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
