# Peer review of "Incorporating Heterogeneous Features into the Random Subspace Method for Bearing Fault Diagnosis"

_entropy, 2023, doi:10.3390/e25081194_

Round 1

Reviewer 1 Report

The paper presents a detailed and well-explained proposal for a bearing diagnosis methodology.  Except for the details that will be referred to below, the article is well structured and well explained, and with some defects corrected, it is publishable. However, this reviewer would like to point out that it is just another work of application of a mixture of different signal processing and machine learning techniques to bearing diagnosis, being one more of the many already published, without being a great contribution to the state of the art of the subject.

Particular comments

- One of the main novelties argued by the authors of this work is the use of the Lasso technique for bearing diagnosis. However, this technique has already been used for this purpose (at least, one ublished in “Energies” and another one in Sdemped17 conefrence), so the authors should at least cite the corresponding references in the text and, if possible, also comment on them and comment on the differences with respect to the work they present.

- A very important part of a methodology such as the one described is the selection of the features to be used for the diagnosis. However, in section 2.3.1, regarding the features in the time domain, the text simply mentions 5 of them and adds “etc.”, defining all of them in a table. Given the importance of this aspect, this would require a much more detailed description. The same comment applies to section 2.3.2 concerning frequency domain features.

- This work uses, as in many other published papers, data from a repository. Although no experimental work has been carried out, at least some commentary on the possibilities of applying this technique in real cases should be included, since the use of vibration analysis has certain practical limitations.

- For the evaluation of the performance of the diagnostic techniques only one ratio is used, it would be interesting to add some more or at least include a confusion matrix of some of the cases analyzed.

- In the references section, citation 12 is incomplete.

Reviewer 2 Report

Incorporating Heterogeneous Features to Random Subspace Method for Bearing Fault diagnosis

1. There are some typos.

2. The proposed algorithm is composed of many methods, and comparing each method extracts the feature. What makes the author can choose the more reliable feature from a specific method? How to compare the methods?

3. Learning based approach is the dataset is important. Is it possible to apply all faults only using the dataset authors used for training?

4. The proposed algorithm needed many steps in the process. In that case, can the algorithm detect a fault in real time? And is it applicable to multiple faults?

5. The Ref. 23 is strange.

6. The reference format is not unified.

Round 2

Reviewer 1 Report

A significantly improved version of the manuscript has been presented and the questions and suggestions of the first revision have been satisfactorily answered.

No comments

Reviewer 2 Report

1.      I think the response in the attached file describes my comments well. However, it is hard to think the revised version is reflected by the response well.

Round 3

Reviewer 2 Report

I think the response in the attached file describes my comments well. There are only minor things. Maybe because the font style differs between the general sentence and mathematical symbols, some symbols can confuse readers. I hope the authors would check and confirm them.

1.      It looks like there are some typos, especially mathematical symbols.

2.      In line 310, what is y_is?
